# Associations of the TyG index with albuminuria and chronic kidney disease in patients with type 2 diabetes

Xiaonan Li, Yuehui Wang *

Department of Geriatrics, Jilin Geriatrics Clinical Research Center, The First Hospital of Jilin University, Changchun, China

* yuehuiwang300@jlu.edu.cn

**Data Availability Statement:** Our data came from the Diabetes Complications Data Set which were collected by the General Hospital of the People's Liberation Army of China from 01/2013 to 12/2017 and it was stored in the Population Health Data

## Abstract

### Objective

Diabetes-related kidney disease reduces patients' quality of life, increases the risk of death, and is associated with insulin resistance (IR). The triglyceride-glucose (TyG) index is a simple and inexpensive alternative to IR measurement. Furthermore, the relationship between albuminuria and chronic kidney disease (CKD) in type 2 diabetes mellitus (T2DM) remains unclear. Therefore, we aimed to investigate the association of TyG index with albuminuria and CKD in patients with T2DM.

### Methods

Data from 01/2013-12/2017 period were obtained from the Population Health Data Archive's Diabetes Complications Data Set. A total of 1048 patients with T2DM were included in this study. CKD is defined as an estimated glomerular filtration rate < 60 ml/min$^{-1}$.1.73 m$^{-2}$ or a urinary albumin-to-creatinine ratio (UACR) $\geq$ 30 mg/g. Albuminuria is defined as a UACR $\geq$ 30 mg/g. The TyG index is calculated by measuring the triglyceride and fasting blood glucose levels. Logistic regression models were used to analyze the association between albuminuria, CKD with T2DM and TyG index.

### Results

We identified 1048 subjects, 63.03% of whom were men. The mean age was 46.21 years, and the mean body mass index was 26.742 kg/m$^2$. CKD and albuminuria detection rates showed an increasing trend in the different TyG subgroups. ($p = 0.008$, $p = 0.006$). Using the Q1 group as a baseline, the risk of albuminuria and CKD was significantly greater in the group Q3 (OR = 1.514, 95% CI 1.121–2.047 $P = 0.05$), and the same result was obtained after adjusting for covariates (OR = 2.241, 95% CI 1.245–4.034, $P = 0.007$). Subgroup analyses revealed a significant increase in the incidence of albuminuria and CKD in the group Q3 compared to that in the Q1 group.

Archive, the website is https://www.ncmi.cn/phda/dataDetails.do?id=CSTR:A0006.11.A0005.201905.000282 (Language is Chinese) The data set contains 87 variables for 3,000 patients with T2DM. The example data from the Diabetes Complications Data Set can be acquired and was uploaded as S1_Table.Once the application has been approved, the data can be viewed on the virtual platform, with specific operational usage uploaded as a support information file named S3_file (Language in Chinese). The username is nmdcfx049 and password is fx0514.lxn.

**Funding:** The author(s) received no specific funding for this work.

**Competing interests:** The authors have declared that no competing interests exist.

## Conclusions

The TyG index is positively associated with albuminuria and CKD in patients with T2DM and may be a marker for predicting the occurrence of early kidney injury in patients with T2DM. Clinicians should test this indicator early to detect lesions and improve patient prognosis.

## Introduction

Type 2 diabetes mellitus (T2DM) is the most common metabolic disorder that leads to numerous microvascular and macrovascular complications [1]. Chronic kidney disease (CKD), a major microvascular complication of T2DM, affects nearly half of the patients with T2DM, and approximately 30% of these patients eventually progress to end-stage renal disease (ESRD) [2–5]. CKD is defined as an estimated glomerular filtration rate (eGFR) < 60 ml/min$^{-1}$.1.73 m$^{-2}$ or increased levels of urinary albumin-to-creatinine ratio (UACR $\geq$ 30 mg/g) [6, 7]. T2DM is a major cause of progression to end-stage renal disease (ESRD). Diabetes-related ESRD severely affects the quality of life and treatment of patients, imposing a significant economic burden [8]. Therefore, delaying progression to ESRD is a major challenge in the management of patients with T2DM.

Insulin resistance (IR), which is characterized by the inability of cells to respond to the action of insulin, is a prominent feature of T2DM and is significantly associated with various diseases, including cardiovascular disease, CKD, and cognitive impairment [9–11]. Studies have shown that IR may be related to increased glomerular hydrostatic pressure, resulting in increased renal vascular permeability, eventually leading to glomerular hyperfiltration [8]. Another hypothesis is that metabolic changes associated with IR cause glomerulosclerosis, glomerular hypertrophy, tubulointerstitial inflammation, and fibrosis, ultimately leading to albuminuria and accelerated CKD progression [12, 13]. Therefore, assessing the IR is crucial for understanding the occurrence and progression of T2DM-related kidney diseases.

Studies have indicated that the TyG index, calculated by combining fasting glucose and triglycerides, has a strong correlation with IR, as evaluated by the Homeostatic Model Assessment for Insulin Resistance (HOMA-IR) and the Hyperinsulinemic-Euglycemic Clamp (HIEC) test [14–16]. Unlike the traditional IR assessment methods (HOMA-IR and HIEC), the TyG index is cost-effective and readily accessible. Multiple studies have demonstrated that the performance of the TyG index in the assessment of IR is consistent with or superior to that of HOMA-IR [17–19]. Thus, the TyG index is emerging as a novel and promising alternative marker for insulin resistance.

Numerous studies have demonstrated a close relationship between the TyG index, T2DM, and vascular lesions [20–24]. Wang L et al. reported that the TyG index, which is independent of known cardiovascular risk factors, can serve as an independent risk factor for predicting the occurrence of major adverse cardiovascular events in patients with T2DM [22]. Moreover, IR has been consistently associated with increased urinary albumin excretion. A cross-sectional study involving 9,872 participants revealed that the TyG index was closely related to albuminuria in the American population [25]. A large observational population-based cohort study conducted by Fritz et al. revealed that the TyG index was independently associated with an increased risk of end-stage renal disease (ESRD), and nearly half of the association between BMI and ESRD risk of ESRD is mediated by the TyG index [26]. Similarly, in a recent cohort study on middle-aged white men conducted by KUNUTSOR S K et al., a higher TyG index was associated with an increased risk of CKD [27]. However, there are few studies on the

relationship between the TyG index, albuminuria, and CKD in individuals with T2DM. Therefore, in patients with a high incidence of T2DM, CKD, and albuminuria, clinicians should perform early TyG assessments to detect early kidney damage so that proactive interventions can be performed. This may reduce the incidence of chronic kidney disease and albuminuria, improve the prognosis, and minimize the risk of chronic complications associated with diabetes. Therefore, we aimed to analyze the relationship between the TyG index and the risk of albuminuria and CKD in patients with T2DM.

## Materials and methods

### Study population

The data come from the Diabetes Complications Data Set which were collected by the General Hospital of the People's Liberation Army of China from 01/2013 to 12/2017 and it was stored in the Population Health Data Archive, The website is https://www.ncmi.cn/phda/dataDetails.do?id=CSTR:A0006.11.A0005.201905.000282 (Language is Chinese) The dataset contains 87 variables for 3,000 patients with T2DM [28]. The example data from the Diabetes Complications Data Set can be acquired and was uploaded as S1 Table. Once the application has been approved, the data can be viewed on the virtual platform, with specific operational usage uploaded as a support information file named S3 File. A secondary analysis of the data from this data set was performed. The study was a retrospective study and conducted using a publicly available database, the requirement of ethical approval for this was waived by the Institutional Review Board of the First Hospital of Jilin University. The need for written informed consent was waived by the Institutional Review Board of the First Hospital of Jilin University due to retrospective nature of the study.

A total of 3000 participants were included in the dataset. After identifying the diagnostic and exclusion criteria, 1048 participants were included. A flowchart of the study population screening is shown in Fig 1.

### Data collection and definitions

The demographic information of the patients, including age, sex, height, weight, blood pressure, history, and medical history, was obtained from the acquired dataset. Laboratory tests: Laboratory indices collected in this study included fasting blood glucose (FPG), glycosylated hemoglobin (HbA1c), total protein (TP), albumin (Alb), total cholesterol (TC), triglyceride (TG), high-density lipoprotein (HDL), low-density lipoprotein (LDL), creatinine, gamma-glutamyl transferase (GGT), uric acid, alanine aminotransferase (ALT), aspartate aminotransferase (AST), lactate dehydrogenase (LDH), alkaline phosphatase (ALP) levels, and the urine albumin/creatinine ratio (UACR).

The TyG index formula is as follows: $\ln$ [FPG (mg/dl) $\times$ TG (mg/dl]). According to TyG tertiles, TyG $\leq$ 8.849 was the Q1 (350) group, TyG 8.849–9.538 was the Q2 (349) group, and TyG > 9.538 was the Q3 (349) group.

eGFR is calculated using the simplified Modification of Diet for Renal Disease (MDRM) formula: eGFR (ml/min$^{-1}$.1.73 m$^{-2}$) = 186 $\times$ serum creatinine$^{-1.154 \times \text{age}-0.203(}\times$0.724 for women).

The definition of albuminuria is a urinary albumin/creatinine ratio $\geq$ 30 mg/g.

DKD is defined as a DKD diagnosis when the study subject meets one or both of the following criteria: (1) an eGFR < 60 ml/min$^{-1}$.1.73 m$^{-2}$ or a UACR $\geq$ 30 mg/g.

Definition of glycemic control: Based on HbA1c, the participants were divided into groups with good glycemic control (< 7%) and groups with poor glycemic control ($\geq$ 7%).

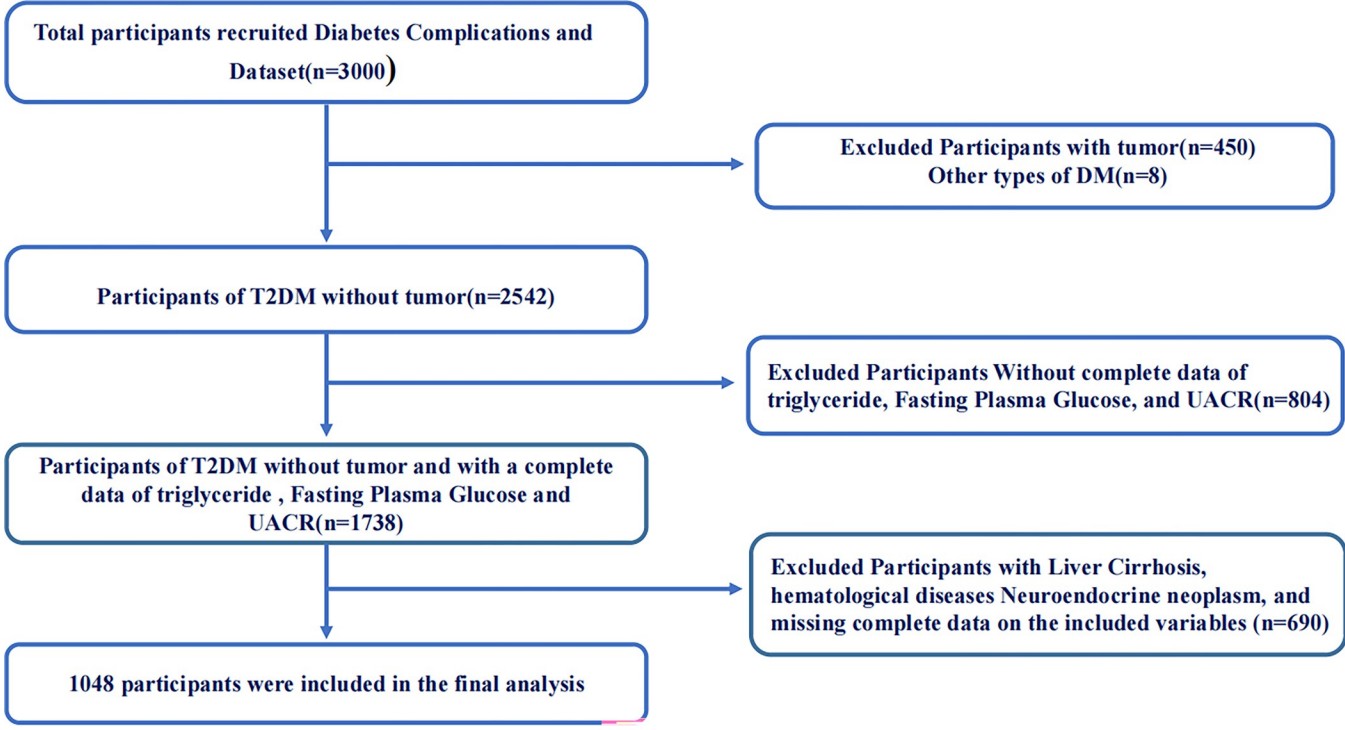

**Fig 1. The flowchart of the study population screening.**

## Statistical analysis

Data were analyzed using SPSS 25.0. Variables with an approximately normal distribution are expressed as the mean ± standard deviation. Variables with a non-normal distribution were presented as medians (interquartile ranges). The t-test or Mann–Whitney U test was used for group comparisons of continuous variables. Categorical variables were described as frequencies or percentages, and the chi-square test was used for group comparisons. Multivariate logistic regression models were used to calculate the odds ratio (OR) and 95% confidence intervals (95% CIs) to clarify the correlation between the TyG index and T2DM with albuminuria and CKD. Further adjustments for covariates and stratified analyses of variables were performed to validate the results. Differences were considered statistically significant at a two-sided significance level of $P< 0.05$.

## Results

### Baseline characteristics of participants

A total of 1048 participants met the inclusion criteria and were included in the analysis. The mean age was 46.21 years, mean body mass index (BMI) was 26.742 kg/m2, and 63.03% of the participants were men. Baseline participant characteristics stratified by the TyG index are shown in Table 1. Compared to those in group Q1, albuminuria, CKD, poor glycemic control, retinopathy, hyperuricemia, and fatty liver were more common in group Q3 ($P < 0.05$). Significant differences were detected in the levels of FPG, HBA1c, albumin, TG, TC, HDL, BUN, creatinine, UA, GGT, and ALP between the three groups ($P < 0.05$). Pairwise comparisons among groups Q3, Q1, and Q2 revealed statistically significant differences in FPG and HBA1c.

Table 1. Baseline characteristics of the participants with different TyG subtypes.

| Variables | Q1(350) | Q2(349) | Q3(349) | F/H/$\chi^2$ | P |
|---|---|---|---|---|---|
| Age (years, $\bar{x}\pm s$) | 55.89±10.759 | 56.42±10.762 | 55.32±12.128 | 0.838 | 0.433 |
| Height (cm, $\bar{x}\pm s$) | 166.541±7.865 | 166.162±7.164 | 166.342±7.940 | 0.179 | 0.836 |
| Weight (kg, $\bar{x}\pm s$) | 73.580±12.213 | 74.375±11.786 | 73.896±13.400 | 0.347 | 0.707 |
| BMI (kg/cm$^2$, $\bar{x}\pm s$) | 24.496±3.448 | 26.836±3.374 | 26.892±4.090 | 1.069 | 0.344 |
| SBP [mmHgM (Q1, Q3)] | 138(124,150) | 135(122.5,150) | 137(124,150) | 1.038 | 0.375 |
| DBP [mmHgM(Q1, Q3)] | 80(73,88) | 80(72,90) | 80(73,90) | 0.887 | 0.447 |
| FBG(mmol/L, $\bar{x}\pm s$) | 5.348±0.820[a] | 7.874±0.808[ab] | 12.817±3.747[abc] | 985.548 | 0.000 |
| HbA1c(%,x±s) | 7.488±1.589[a] | 8.103±1.468[ab] | 9.493±1.772[abc] | 141.403 | 0.000 |
| TP(g/L, $\bar{x}\pm s$)) | 64.838±7.235 | 66.030±5.998a | 65.565±6.661 | 3.514 | 0.173 |
| Alb(g/L, $\bar{x}\pm s$)) | 40.103±5.573 | 40.981±4.641[a] | 40.134±4.983[b] | 9.042 | 0.011 |
| TG [mmol/L, M (Q1, Q3)] | 1.450(1.020,2.020) | 1.740(1.170,2.440) [a] | 1.810(1.260,2.65) [a] | 34.118 | 0.000 |
| TC [mmol/L, M (Q1, Q3)] | 4.335(3.600,5.125) | 4.540(3.910,5.275) [a] | 4.590(3.900,5.440) [a] | 11.705 | 0.000 |
| HDL [mmol/L, M (Q1, Q3)] | 1.050(0.907,1.230) | 0.990(0.850,1.145) [a] | 1.000(0.850,1.200) [a] | 10.987 | 0.004 |
| LDH [mmol/L, M (Q1, Q3)] | 2.705(2.060,3.360) | 2.790(2.310,3.375) | 2.820(2.250,3.475) | 4.665 | 0.097 |
| BUN [umol/L, M (Q1, Q3)] | 5.530(4.580,6.707) | 5.420(4.325,6.535) | 5.450(4.510,6.675)[a] | 2.355 | 0.038 |
| Creatinine [umol/L, M (Q1, Q3)] | 73.550(60.850,89.000) | 69.100(56.000,89.000) [a] | 65.000(54.250,82.300) [a] | 24.025 | 0.000 |
| Uric Acid [umol/L, M (Q1, Q3)] | 325.300(267.775,388.275) | 311.300(265.100,386.800) | 300.700(246.950,366.150) [ab] | 11.188 | 0.004 |
| ALT [U/L, M (Q1, Q3)] | 17.100(12.850,25.65) | 18.200(13.250,28.200) | 18.300(13.000,26.400) | 1.713 | 0.425 |
| AST [U/L, M (Q1, Q3)] | 15.950(13.500,19.700) | 16.100(12.950,21.450) | 16.30(13.10,22.45) | 1.624 | 0.444 |
| GGT [U/L, M (Q1, Q3)] | 22.400(15.750,35.225) | 24.200(16.450,41.000) [a] | 24.900(17.100,43.650) [a] | 7.076 | 0.029 |
| ALP [U/L, M (Q1, Q3)] | 60.600(50.700,71.300) | 62.850(53.380,77.475) [a] | 71.000(59.650,85.350) [ab] | 39.793 | 0.000 |
| UACR [mg/g, M (Q1, Q3)] | 13.500(7.000,146.250) | 17.000(7.000,92.500) | 28(10.000,186.500) [ab] | 16.794 | 0.000 |
| Male gender[n,(%)] | 233(35.1) | 218(32.9) | 212(32.0) | 2.696 | 0.260 |
| Hypertension[n,(%)] | 234(34.7) | 219(32.50) | 221(32.8) | 1.507 | 0.471 |
| Hyperlipidemia[n,(%)] | 62(30.8) | 64(31.8) | 75(37.3) | 1.846 | 0.397 |
| Atherosclerosis[n,(%)] | 168(31.0) | 183(33.8) | 191(35.0) | 3.276 | 0.194 |
| Coronary heart disease[n,(%)] | 13(33.8) | 85(32.2) | 91(34.5) | 0.274 | 0.872 |
| Hyperuricemia[n,(%)] | 73(33.8) | 86(39.80) | 57(26.4) [a] | 7.383 | 0.025 |
| Age>60 Years[n,(%)] | 137(34.20) | 136(33.90) | 128(37.90) | 0.560 | 0.756 |
| Fatty liver [n,(%)] | 128(27.3) | 160(34.1) [a] | 181(38.6)[a] | 22.948 | 0.000 |
| CKD] [(n,(%)] | 138(30.7) | 138(30.7) | 173(38.50) [ab] | 9.669 | 0.008 |
| Albuminuria [n,(%)] | 133(30.8) | 131(30.3) | 168(38.9) [ab] | 10.345 | 0.006 |
| HbA1c%>7[n,(%)] | 183(23.9) | 261(34.1) [a] | 321(42.1) [ab] | 140.513 | 0.000 |
| Retinopathy[n,(%)] | 222(30.50) | 238(32.7) | 268(36.8) [ab] | 15.109 | 0.001 |

SD, standard deviation; M, median; Q1:1st quartile, Q3:3rd quartile, TYG, triglyceride-glucose product index; HbA1c, glycated hemoglobin; TG, triglyceride; TC, total cholesterol; HDL-C, high-density lipoprotein; LDL-C, low-density lipoprotein; eGFR, glomerular filtration rate; UACR, urinary albumin-to-creatinine ratio; CKD, chronic kidney disease. Statistical significance was set at $p < 0.05$. difference. a $p<0.05$ compared with Group Q1, b indicates $p<0.05$ compared with Group Q2, c $p<0.05$, compared with Group Q2.

## Comparison of the proportions of patients with albuminuria and CKD detected in TyG index tertile subgroups

The detection rates of CKD and albuminuria in the included population were 42.8% (449/1048), and 41.2% (432/1048). Compared to the incidence rates in Group Q1, the incidence of CKD and albuminuria in Group Q3 significantly increased ($\chi^2$ = 9.669, $P < 0.05$; $\chi^2$ = 10.345, $P < 0.05$). The results are presented both in Fig 2 and Table 2.

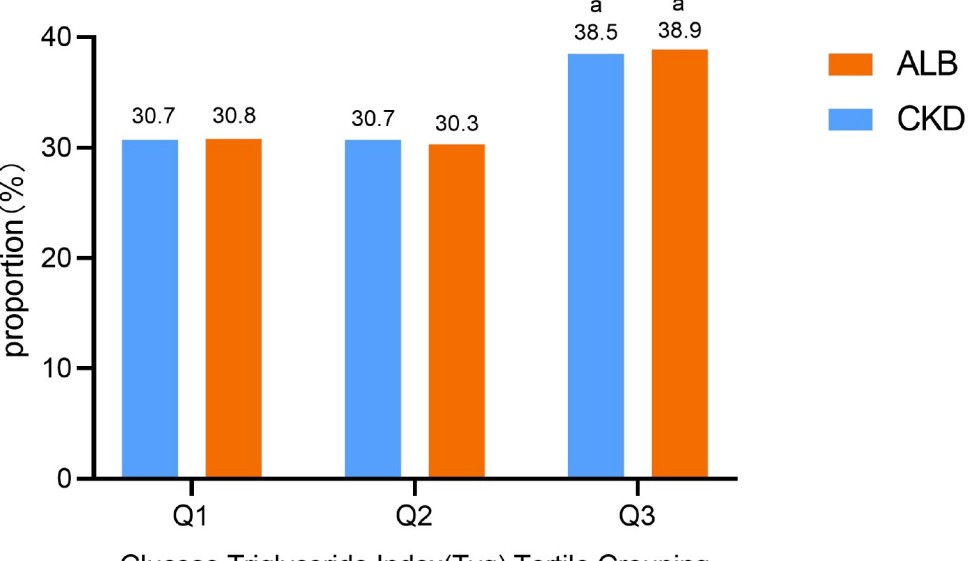

**Fig 2. The proportions of albuminuria and CKD detected in TyG index tertile subgroups.** Compared to the incidence rates in Group Q1, the incidence of CKD and albuminuria in Group Q3 significantly increased. ALB: albuminuria; CKD: chronic kidney disease.

## Association of the risk of albuminuria in different TyG subgroups

In this study, we applied two models to investigate the relationship between TyG index and albuminuria. To analyze the results, we used the TyG tertiles as subgroups and albuminuria as the dependent variable. From Table 3 we can see that according to the unadjusted covariate factors of Model 1, the risk of albuminuria was significantly greater in Group Q3 than in Group Q1 (OR = 1.514, 95% CI 1.121–2.047; $p$ = 0.007). According to Model 2, after adjusting for the covariates of hypertension, fatty liver, hyperlipidemia, coronary heart disease, HbA1c,

**Table 2. Comparison of the prevalence of albuminuria and CKD among TYG tertile subgroups.**

| Clinical features | Q1 | Q2 | Q3 | $\chi^2$ | P |
|---|---|---|---|---|---|
| CKD | 138(30.7) | 138(30.7) | 173(38.50) [a] | 9.669 | 0.008 |
| Albuminuria | 133(30.8) | 131(30.3) | 168(38.9) [a] | 10.345 | 0.006 |

a Comparison with the Q1 group. $p < 0.05$ indicates a statistically significant difference

**Table 3. Logistic regression analysis of albuminuria risk in different TyG subgroups.**

| Variables | Model 1 | | Model 2 | |
|---|---|---|---|---|
| | OR(95%CI) | P value | OR(95%CI) | P value |
| TyG | | | | |
| Q1 | 1 | | 1 | |
| Q2 | 0.980 (0.722–1.331) | 0.899 | 1.270 (0.863–1.868) | 0.255 |
| Q3 | 1.514 (1.121–2.047) | 0.007 | 2.241 (1.245–4.034) | 0.007 |

Model 1: Uncorrected for confounders; Model 2: Corrected for hypertension, fatty liver, hyperlipidemia, coronary heart disease, total cholesterol, glycosylated hemoglobin, sex, age, fasting glucose, retinopathy, and eGFR.

**Table 4. Association of the TyG index with albuminuria in populations stratified by different variables.**

| Variables Layers | Numbers | TYG Q1 | TYG Q2 OR(95%CI) | P value | TYG Q3 OR(95%CI) | P value |
|---|---|---|---|---|---|---|
| Gender | | | | | | |
| Male | 663 | | | 0.203 | | 0.001 |
| Female | 385 | | 1.475 (0.810–2.686) | 0.563 | 2.750 (1.505–5.027) | 0.000 |
| Age(years) | | | | | | |
| ≤60 | 647 | | 1.190 (0.766–1.851) | 0.700 | 2.066 (1.317–3.241) | 0.001 |
| >60 | 401 | | 1.999 (1.174–3.402) | 0.339 | 1.658 (0.949–2.896) | 0.001 |
| HbA1c (%) | | | | | | |
| <7 | 283. | | 1.084 (0.587–2.004) | 0.796 | 3.189 (1.278–7.957) | 0.013 |
| ≥7 | 765. | | 1.143 (0.749–1.746) | 0.536 | 1.975 (1.309–2.978) | 0.001 |
| Hypertension | | | | | | |
| yes | 674 | | 1.199 (0.803–1.790) | 0.374 | 2.079 (1.376–3.140) | 0.001 |
| no | 374 | | 1.384 (0.722–2.654) | 0.328 | 3.288 (1.735–6.233) | 0.000 |
| Fatty liver | | | | | | |
| yes | 469 | | 0.960 (0.566–1.627) | 0.879 | 0.960 (0.566–1.627) | 0.023 |
| no | 579 | | 1.504 (0.966–2.34.) | 0.071 | 2.825 (1.772–4.503) | 0.000 |

sex, age, FPG level, retinopathy, and eGFR, the incidence of albuminuria was 2.241 times greater in group Q3 than in group Q1 (OR = 2.241, 95% CI 1.245–4.034; P = 0.007). Stratified analysis of the variables was conducted to verify the reliability of the results. Table 4 shows the results presented after stratification. After adjusting for other covariate factors, compared to incidence rates in Group Q1, the incidence of albuminuria in Group Q3 significantly increased across different categories, including sex (male, female), age (≤ 60 years, > 60 years), HbA1c (< 7%, ≥ 7%), hypertension (yes or no), and fatty liver (yes or no) (Table 4). In summary, TyG index is an independent predictor of albuminuria in patients with T2DM.

## Associations between the risk of CKD and different TyG subgroups

To further investigate the relationship between TyG index and CKD, we used two models. In Model 1, which was not adjusted for covariates and was consistent with the urinary albumin results, compared with that in the Q1 group, the risk of CKD was significantly higher in the group Q3 (OR = 1.510, 95% CI 1.119–2.038, P = 0.007). Model 2 was adjusted for covariates, and after adjustment for hypertension, fatty liver, hyperlipidemia, coronary heart disease, HbA1c, sex, age, FPG, retinopathy, and eGFR, a 1.947-fold greater frequency of CKD was detected in the group Q3 than in the Q1 group (OR = 1.947, 95% CI 1.068–3.548, P = 0.030), we can see the results in Table 5. To validate the relationship between the TyG index and

**Table 5. Logistic regression analysis of CKD risk in different TyG subgroups.**

| Variable | Model 1 OR(95%CI) | P value | Model 2 OR(95%CI) | P value |
|---|---|---|---|---|
| TyG | | | | |
| Q1 | 1 | | 1 | |
| Q2 | 1.005 (0.742–1.361) | 0.976 | 1.254 (0.851–1.849) | 0.252 |
| Q3 | 1.510 (1.119–2.038) | 0.007 | 1.947 (1.068–3.548) | 0.030 |

Model 1: Unadjusted for covariates. Model 2: Adjusted for hypertension, fatty liver, stroke, hyperlipidemia, coronary heart disease, total cholesterol, HDL cholesterol, glycosylated hemoglobin subgroups, and sex.

**Table 6. Association of the TyG index with CKD in populations stratified by different variables.**

| Variables and Layers | numbers | TYG Q1 | TYG Q2 | | TYG Q3 | |
|---|---|---|---|---|---|---|
| | | | OR(95%CI) | P value | OR(95%CI) | P value |
| gender | | | | | | |
| male | 663 | | 1.170 (0.770–1.778) | 0.461 | 2.523 (1.630–3.907) | 0.000 |
| Female | 385 | | 1.582 (0.867–2.886) | 0.135 | 2.798 (1.520–5.149) | 0.001 |
| Age | | | | | | |
| ≤60 | 647 | | 1.140 (0.741–1.753) | 0.551 | 2.345 (1.506–3.650) | 0.000 |
| >60 | 401 | | 1.315 (0.743–2.327) | 0.347 | 2.725 (1.528–4.858) | 0.001 |
| HbA1c | | | | | | |
| <7 | 283 | | 1.239 (0.668–2.298) | 0.497 | 5.043 (1.904–13.352) | 0.001 |
| ≥7 | 765 | | 1.171 (0.764–1.795) | 0.469 | 2.020 (1.332–3.064)) | 0.001 |
| Hyperuricemia | | | | | | |
| yes | 216 | | 1.799 (0.872–3.709) | 0.112 | 2.434 (1.044–5.677) | 0.040 |
| no | 832 | | 1.139 (0.770–1.684) | 0.515 | 2.454 (1.668–3.612) | 0.000 |
| Hypertension | | | | | | |
| yes | 674 | | 1.309 (0.873–1.965) | 0.194 | 2.241 (1.469–3.418) | 0.000 |
| no | 374 | | 1.335 (0.698–2.555) | 0.382 | 3.404 (1.799–6.440) | 0.000 |
| Fatty liver | | | | | | |
| yes | 469 | | 1.086 (0.640–1.842) | 0.760 | 1.922 (1.143–3.232) | 0.014 |
| no | 579 | | 1.504 (0.960–2.357) | 0.075 | 3.189 (1.972–5.157) | 0.000 |

CKD, we conducted stratified analyses of the variables. After adjusting for covariates, the results showed that among patients categorized by sex (male, female), age ($\leq 60$ years, $> 60$ years), HbA1c ($< 7\%$, $\geq 7\%$), hypertension (yes or no), and fatty liver (yes or no), the incidence of CKD in group Q3 was significantly greater than that in group Q1, Table 6 describes this result. Similarly, TyG index is an independent predictor of CKD in patients with T2DM.

## Discussion

This study investigated the potential correlation between TyG index and the presence of albuminuria and CKD in patients with T2DM. In this single-center retrospective cohort study, we found a strong positive correlation between the TyG index, albuminuria, and CKD with T2DM. After adjusting for multiple covariates, the positive correlation remained stable, suggesting that baseline TyG index is an independent predictor of albuminuria and CKD in patients with T2DM. This association remained consistent across subgroups of sex, age, HbA1c level, hypertension, fatty liver, and hyperuricemia.

IR, an important feature of T2DM, is particularly associated with nephropathy and a decline in kidney function and is present in the early stages of CKD [29–31]. Albuminuria, a marker of early kidney damage, is closely associated with IR [32, 33]. PAN Y et al. reported that patients with an elevated TyG index have a greater risk of developing albuminuria, which is consistent with our results and is more pronounced in patients with poor glycemic control [34]. There are several possible mechanisms through which IR may be associated with kidney damage. The kidney contains a variety of insulin-sensitive cells that express insulin receptors [35]. When IR occurs, insulin signaling is impaired and glomerular filtration pressure is increased, leading to glomerular hyperfiltration [36], cytoskeletal rearrangement [37], mitochondrial dysfunction [38] and inflammation [35], resulting in renal hemodynamics, disruption of podocyte viability, and accelerated progression of CKD by facilitating glomerular

capillary dilation and tubular fibrosis [12, 13] Therefore, early detection of IR is crucial for the prevention of kidney disease.

Currently, the gold standard for assessing IR is the HIEC test; however, it is invasive and expensive, used only for academic research, and not applied in clinical practice [39]. The widely used HOMA-IR index, which assesses β-cell function and insulin resistance, has limited value in patients undergoing insulin therapy or those without functional β-cells, and serum insulin measurement is not widely used in clinical practice [39]. The TyG index, calculated using FPG and TG levels, assesses IR in both diabetic and non-diabetic individuals and is equivalent to or even better than HOMA-IR [14–16, 40]. This alternative method is simple, convenient, and low-cost, does not require insulin quantification, and can be used in all patients regardless of their insulin treatment status [41]. Therefore, in our study, we adopted the TyG index as an alternative biomarker for IR because it is readily accessible in routine clinical practice and widely used.

The TyG index, a new surrogate for insulin resistance measurement, is increasingly supported by evidence of its important role in predicting vascular diseases [42, 43] A meta-analysis indicated that the TyG index is positively associated with the incidence rates of heart failure and adverse outcomes. Similarly, a study also suggests that the prevalence of other common cardiovascular and cerebrovascular diseases, including arteriosclerosis, coronary artery calcification, coronary artery disease, myocardial infarction, and atrial fibrillation, is significantly related to a high TyG index [44–50]. However, the association between the TyG index and renal microvascular damage was low. Our study showed that an elevated TyG index is significantly associated with the risk of albumin deficiency and CKD in patients with T2DM. A high TyG index has been shown to exacerbate renal dysfunction by promoting insulin resistance and related metabolic disturbances [50] and it is a reliable and convenient prognostic marker for adverse outcomes in patients with kidney disease [51]. A cohort study based on CHARLS data revealed an increased risk of CKD in the highest quartile compared to that in the lowest quartile [52], which is the same as that found in our study. Interestingly, Viñas et al. conducted basic experiments to demonstrate that female mice were more resistant to kidney injury than male mice [50, 53]. Saliba et al. reported the same result in humans, which may be related to the estrogen-AMPK pathway [54]. However, in our study, we did not find any difference in albumin and CKD levels by sex, even after adjusting for risk factors and sex for stratification; this may be related to the ethnicity of the included population, geographic region, and the number of patients included, which needs to be further explored.

Our study had several limitations. First, it was retrospective, precluding the establishment of a causal relationship between our variables of interest and outcomes, and there may have been reporting and selection biases. Second, although potential confounders were controlled as covariates in the multivariate regression model, it is important to acknowledge that the impact of unrecorded confounders cannot be categorically excluded. Finally, our study was a single-center study with a moderate sample size; future research should include large-scale multicenter longitudinal and prospective studies to further investigate the effects and mechanisms of the TyG index in T2DM patients with albuminuria and CKD.

## Conclusions

In conclusion, this study revealed that the TyG index was positively associated with CKD and albuminuria in patients with T2DM. Our findings have clinical and public health significance because of the increasing prevalence of diabetes and growing population of patients with diabetes-related kidney injuries. Clinicians should perform early testing of the TyG index to identify early kidney damage, aggressively intervene to reduce proteinuria and CKD, and improve prognosis, thereby reducing the risk of chronic complications of diabetes mellitus.

## Supporting information

**S1 Table. The example data from the diabetes complications data set.**
(XLSX)

**S1 File. Human participants research checklist.**
(PDF)

**S2 File. Certificate_of_editing.**
(PDF)

**S3 File. Virtual desktop user instructions.**
(PDF)

## Acknowledgments

We are thankful for data from the Population Health Data Archive.

## Author Contributions

**Conceptualization:** Yuehui Wang.

**Data curation:** Xiaonan Li.

**Formal analysis:** Xiaonan Li.

**Methodology:** Xiaonan Li.

**Project administration:** Yuehui Wang.

**Software:** Xiaonan Li.

**Supervision:** Yuehui Wang.

**Visualization:** Yuehui Wang.

**Writing – original draft:** Xiaonan Li.

**Writing – review & editing:** Yuehui Wang.

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
