## [Decision Letter · Decision Letter 0]

9 May 2024

PONE-D-24-07664Association of TyG index with albuminuria and Chronic Kidney Disease with Type 2 Diabetes MellitusPLOS ONE

Dear Dr. wang,

Thank you for submitting your manuscript to PLOS ONE. After careful consideration, we feel that it has merit but does not fully meet PLOS ONE’s publication criteria as it currently stands. Therefore, we invite you to submit a revised version of the manuscript that addresses the points raised during the review process.

We look forward to receiving your revised manuscript.

Kind regards,

Amirmohammad Khalaji

Academic Editor

PLOS ONE

“no”

3. For studies involving third-party data, we encourage authors to share any data specific to their analyses that they can legally distribute. PLOS recognizes, however, that authors may be using third-party data they do not have the rights to share. When third-party data cannot be publicly shared, authors must provide all information necessary for interested researchers to apply to gain access to the data. (https://journals.plos.org/plosone/s/data-availability#loc-acceptable-data-access-restrictions)

a) A description of the data set and the third-party source

b) If applicable, verification of permission to use the data set

c) Confirmation of whether the authors received any special privileges in accessing the data that other researchers would not have

d) All necessary contact information others would need to apply to gain access to the data

Reviewers' comments:

Reviewer's Responses to Questions

**Comments to the Author**

1. Is the manuscript technically sound, and do the data support the conclusions?

Reviewer #1: No

Reviewer #2: Yes

2. Has the statistical analysis been performed appropriately and rigorously? 

Reviewer #1: I Don't Know

Reviewer #2: Yes

3. Have the authors made all data underlying the findings in their manuscript fully available?

Reviewer #1: No

Reviewer #2: No

4. Is the manuscript presented in an intelligible fashion and written in standard English?

Reviewer #1: No

Reviewer #2: Yes

5. Review Comments to the Author

Reviewer #1: This manuscript examined the connection between the triglyceride-glucose (TyG) index, albuminuria, and chronic kidney disease (CKD) in individuals with type 2 diabetes mellitus (T2DM). They analyzed data from 3000 patients, of whom 1048 met the study's criteria. CKD was determined by a glomerular filtration rate (eGFR) < 60 ml/min-1.73 m-2 or a urinary albumin-to-creatinine ratio (UACR) ≥ 30 mg/g, while albuminuria was defined as UACR ≥ 30 mg/g. Results indicated a positive association between the TyG index and both albuminuria and CKD, with higher odds ratios observed for those in the highest TyG index quartile.

the whole manuscript sections (introduction, method, result, discussion) should be revised. The manuscript does not follow a standard structure of manuscript writing. I recommend a native English-speaking revision to address the many stylistic issues, the manuscript lacks fluency and the English in the text is EXTREMELY poor, complete revision is necessary. these are some examples of the text that should be revised:

Abstract:

1. The objective should be explained comprehensively, pointing to the aim, benefits, and necessity of the study.

2. In the method section, mention the timeline of data acquisition.

3. In the result section, include patient characteristics such as mean age, sex, BMI, etc.

4. In the conclusion, clarify the clinical usage of these findings.

the whole manuscript sections (introduction, method, result, discussion) should be revised. The manuscript does not follow a standard structure of manuscript writing. I recommend a native English-speaking revision to address the many stylistic issues, the manuscript lacks fluency and the English in the text is EXTREMELY poor, complete revision is necessary. these are some examples of the text that should be revised:

introduction:

The results concerning hard end-points should be stated. In my opinion it is too brief and not focused on clinical aspects clarifying the specific factors, tests, or markers evaluated and their significance in understanding the pathological roles in patients with CKD. these are some of the many grammatical, stylistic and spelling errors:

-“Chronic kidney disease (CKD) is a microvascular complication of diabetes mellitus (DM)(1). As the prevalence of CKD increases, so does the risk of cardiovascular disease, end-stage renal disease (ESRD), and cumulative mortality (2)” the whole paragraph should be revised.

- Paraphrase "CKD WAS defined as estimated glomerular filtration …" to "CKD is defined as estimated glomerular filtration…"

-some well-defined and basic scientific facts are mentioned as scientific suggestion � for example: “It has been suggested that insulin resistance (IR) is closely associated with an increased risk of kidney disease in diabetic populations”

-there should be an explanation regarding the definition of TyG index, its utility and previous research on this index. Delve into the concept of clinical importance and the superiority of using of this index in CKD.

Overall, significant revisions are required for structural, grammatical, and clarity issues. Additionally, the clinical implications of the findings need further discussion.

Methods:

-This section does not follow the standard structure of a manuscript and should be paraphrase. For example, is written in multiple title sections for each item which can be written in a paragraph.

-Consider replacing methodology to methods.

Results:

The whole section should be revised.

In the first paragraph of the results section there should be a brief presentation of the patients characteristics (mean age, sex, BMI, etc.) and then point to main findings and after that consider mentioning the tables.

Discussion:

Aside from many scientific errors, this section does not follow the standard structure of a manuscript and should be paraphrase. The topics discussed in the discussion have not been presented in a relevant, orderly and coherent manner. Again, I recommend a native English-speaking revision.

For example, in the first paragraph there should be a brief mention of the main results of the study.

“The homeostasis model assessment of insulin resistance (HOMA-IR) is a widely used index of insulin resistance in clinical practice. However, due to the need for clinical blood collection and the existence of low utility, this index has certain limitations, and many other simplified clinical indicators can be comparable to it, such as the TG/HDL-C ratio, lipid accumulation index, visceral adiposity index, METS-IR, and TyG index. It was shown that the TyG index can be an indicator with good agreement with the HOMA-IR and that it is a more convenient and quicker way of assessing glycolipid metabolism than the HOMA-IR”

- The claim that the TyG index does not require blood collection for its measurement, while HOMA-IR does, is incorrect. Both the TyG index and HOMA-IR require blood collection for their measurements. The TyG index is calculated using fasting triglycerides and fasting plasma glucose, both of which necessitate blood collection. Similarly, HOMA-IR, which assesses insulin resistance, also requires blood collection to measure insulin levels and glucose levels.

After careful evaluation, I regret to inform you that I cannot recommend this article for publication. In my assessment, the manuscript falls short of meeting even the most basic standards expected of a quality manuscript. Several critical issues have been identified across various sections, including but not limited to, inadequate explanation of the objective and significance of the study in the abstract, lack of clarity and coherence in the introduction, methodological deficiencies, insufficient presentation and interpretation of results, and a discussion lacking in relevance and scientific rigor. Given the substantial shortcomings observed throughout the manuscript, I believe that publishing it in its current form would not contribute positively to the scientific literature.

Reviewer #2: Li et al. have performed a study on the association between the TyG index and CKD in patients with T2DM. The manuscript is well-written. These are my comments:

- The authors have mentioned one limitation of the manuscript. Weren’t there any other limitations in terms of methodology, drafting, and interpreting the results?

- A paragraph discussing the clinical applications of these findings and suggestions for further research is needed in the discussion section.

- There are several meta-analysis studies that have assessed the TyG index in several diseases (such as HF doi: 10.1186/s12933-023-01973-7, stroke: 10.1186/s12933-022-01732-0, ACS: 10.1186/s12933-023-01906-4, CVDs in the general population: 10.1186/s12933-022-01546-0, AF: 10.1186/s40001-024-01716-8, arterial stiffness: https://doi.org/10.1186/s12933-023-01819-2). These could be discussed in the discussion as among the applications of this novel index.

6. PLOS authors have the option to publish the peer review history of their article (what does this mean?). If published, this will include your full peer review and any attached files.

Reviewer #1: No

Reviewer #2: No

---

## [Author Response · Author response to Decision Letter 0]

29 May 2024

Manuscript ID: PONE-D-24-07664

Manuscript title: Association of TyG index with albuminuria and Chronic Kidney Disease with Type 2 Diabetes Mellitus

Dear Editor: 

Thank you very much for your attention, evaluation, and comments on our paper "Association of TyG index with albuminuria and Chronic Kidney Disease with Type 2 Diabetes Mellitus". We have revised the manuscript according to your kind advice and the reviewer’s detailed suggestions and tried our best to revise the manuscript. Please find the responses to the reviewers below. We sincerely hope this manuscript will be finally accepted for publication in "PLOS ONE". We would love to thank 

you for allowing us to resubmit a revised copy of the manuscript and we highly appreciate your time and consideration.

The following is the point-by-point response to the editor and reviewers.

Revised portions are marked in purple in the revised paper. The main corrections in the paper and the responses to the editor’s comments are the following

Editor #

Comment 1: Please ensure that your manuscript meets PLOS ONE's style requirements, including those for file naming. The PLOS ONE style templates can be found at https://journals.plos.org/plosone/s/file?id=wjVg/PLOSOne_formatting_sample_main_body.pdf and

Response: Thanks to your prompt attention, we have revised the manuscript according to PLOS ONE's style requirements.

Comment 2. Thank you for stating the following in your Competing Interests section: “no” Please complete your Competing Interests on the online submission form to state any Competing Interests. If you have no competing interests, please state "The authors have declared that no competing interests exist.", as detailed online in our guide for authors at http://journals.plos.org/plosone/s/submit-now.This information should be included in your cover letter; we will change the online submission form on your behalf.

Response: We appreciate your suggestions and we have made the appropriate changes. The statement of “The authors have declared that no competing interests exist” are put in the cover letter and manuscript.

Comment 3. For studies involving third-party data, we encourage authors to share any data specific to their analyses that they can legally distribute. PLOS recognizes, however, that authors may be using third-party data they do not have the right to share. When third-party data cannot be publicly shared, authors must provide all information necessary for interested researchers to apply to gain access to the data. (https://journals.plos.org/plosone/s/data-availability#loc-acceptable-data-access-restrictions)For any third-party data that the authors cannot legally distribute, they should include the following information in their Data Availability Statement upon submission: a) A description of the data set and the third-party sources) If applicable, verification of permission to use the data set. c) Confirmation of whether the authors received any special privileges in accessing the data that other researchers would not haved) All necessary contact information others would need to apply to gain access to the data.

Response: The data for this study were obtained from the Population Health Data Archive's Diabetes Complications Data Set( A Public database) from 01/2013 to 12/2017. Users can request and access data at https://www.ncmi.cn//phda/dataDetails.do?id=CSTR:A0006.11.A0005.201905.000282, applicants can also get information about the application by contacting the email Icyx301@126.com.

Comment 4. PLOS requires an ORCID iD for the corresponseing author in Editorial Manager on papers submitted after December 6th, 2016. Please ensure that you have an ORCID iD and that it is validated in Editorial Manager. To do this, go to ‘Update my Information’ (in the upper left-hand corner of the main menu), and click on the Fetch/Validate link next to the ORCID field. This will take you to the ORCID site and allow you to create a new iD or authenticate a pre-existing iD in Editorial Manager. Please see the following video for instructions on linking an ORCID iD to your Editorial Manager account: https://www.youtube.com/watch?v=_xcclfuvtxQ

Response: Thanks for the advice, I already have an ORCID iD, and it is validated in Editorial Manager.

Comment 5: Please include your full ethics statement in the ‘Methods’ section of your manuscript file. In your statement, please include the full name of the IRB or ethics committee who approved or waived your study, as well as whether or not you obtained informed written or verbal consent. If consent was waived for your study, please include this information in your statement as well.

Response: We appreciate your advice, the study was conducted from a third-party database with identifiable data, and the data on Diabetes Complications Data Set were stored in the Population Health Data Archive, which contains 87 variables for 3,000 patients with T2DM. A secondary analysis of the data from this data set was performed. The study was a retrospective study and conducted using a third-party database with identifiable data; and the research project did not involve personal or commercial interests, the requirement of ethical approval for this was waived by the Institutional Review Board of the First Hospital of Jilin University.The need for written informed consent was waived by the Institutional Review Board of the First Hospital of Jilin University due to retrospective nature of the study.

Revised portions are marked in red in the revised paper. The main corrections in the paper and the response to the reviewer’s comments are as follows

Reviewer #1

Comment 1 This manuscript examined the connection between the triglyceride-glucose (TyG) index, albuminuria, and chronic kidney disease (CKD) in individuals with type 2 diabetes mellitus (T2DM). They analyzed data from 3000 patients, of whom 1048 met the study's criteria. CKD was determined by a glomerular filtration rate (eGFR) < 60 ml/min-1.73 m-2 or a urinary albumin-to-creatinine ratio (UACR) ≥ 30 mg/g, while albuminuria was defined as UACR ≥ 30 mg/g. Results indicated a positive association between the TyG index and both albuminuria and CKD, with higher odds ratios observed for those in the highest TyG index quartile. the whole manuscript sections (introduction, method, result, discussion) should be revised. The manuscript does not follow a standard structure of manuscript writing. I recommend a native English-speaking revision to address the many stylistic issues, the manuscript lacks fluency and the English in the text is EXTREMELY poor, complete revision is necessary. these are some examples of the text that should be revised.

Response: Thank you very much for your suggestions and your kind work. I'm very sorry for the poor English, I've invited a native English speaker to revise the whole manuscript, and we've also revised the article point by point according to your comments on the problems you've raised in the article, so I hope our revisions can make your reading more satisfying.

Abstract

Comment 2: The objective should be explained comprehensively, pointing to the aim, benefits, and necessity of the study.

Response: Thank you for your serious work, We've taken your suggestion and written it up. As the incidence of diabetes mellitus increases, diabetes mellitus-related kidney injury also shows a trend of gradual increase, there is no targeted treatment for diabetic kidney injury, so the prevention is greater than the treatment. Insulin resistance is an important risk factor for diabetic kidney injury, and the TyG index is economical, cheap, and easy to promote the resistance of insulin as an indicator, and it is less explored in the diabetic kidney injury research. Therefore, the purpose of this study is to investigate the relationship between the TyG index and albumin and CKD of T2DM and to search for an excellent index that can provide clinicians with an early evaluation of the disease. You can see it below

Objective: Diabetes-related kidney disease reduces patients' quality of life, increases the risk of death, and is associated with insulin resistance (IR). The triglyceride-glucose (TyG) index is a simple and inexpensive alternative to IR measurement. Furthermore, the relationship between albuminuria and chronic kidney disease (CKD) in type 2 diabetes mellitus (T2DM) remains unclear. Therefore, we aimed to investigate the association of TyG index with albuminuria and CKD in patients with T2DM.

Comment 3: In the method section, mention the timeline of data acquisition.

Response: Thank you for your suggestion. the timeline of data acquisition is from 01/2013 to 12/2017., and we have also added a flowchart (Fig 1) of participant inclusion in the study to more clearly define our process.

Comment 4: In the result section, include patient characteristics such as mean age, sex, BMI, etc.

Response: Thanks for your suggestion, I have described the characteristics of the inclusion population in the results section. The changes to the results section can be seen in the revised text and below.

Results: We identified 1048 subjects, 63.03% of whom were men. The mean age was 46.21 years, and the mean body mass index was 26.742 kg/m2. CKD and albuminuria detection rates showed an increasing trend in the different TyG subgroups. (p = 0.008, p = 0.006). Using the Q1 group as a baseline, the risk of albuminuria and CKD was significantly greater in the group Q3 (OR = 1.514, 95% CI 1.121-2.047 P = 0.05), and the same result was obtained after adjusting for covariates (OR = 2.241, 95% CI 1.245-4.034, P = 0.007). Subgroup analyses revealed a significant increase in the incidence of albuminuria and CKD in the group Q3 compared to that in the Q1 group. 

Comment 5: In the conclusion, clarify the clinical usage of these findings

Response: We are very grateful to the reviewers for your comments, our findings have clinical and public health significance because of the increasing prevalence of diabetes and the growing population with diabetes-related kidney injury and the TyG index is positively associated with albuminuria and CKD in patients with T2DM and may be a marker for predicting the occurrence of early kidney injury in patients with T2DM. So clinicians can perform early testing TyG index to identify early kidney damage and intervene aggressively to reduce proteinuria and CKD and improve prognosis, thereby reducing the risk of chronic complications of diabetes mellitus. we have changed the results section based on your suggestion. We have revised the abstract, the exact description of which can be seen in the revised manuscript and below

Objective: Diabetes-related kidney disease reduces patients' quality of life, increases the risk of death, and is associated with insulin resistance (IR). The triglyceride-glucose (TyG) index is a simple and inexpensive alternative to IR measurement. Furthermore, the relationship between albuminuria and chronic kidney disease (CKD) in type 2 diabetes mellitus (T2DM) remains unclear. Therefore, we aimed to investigate the association of TyG index with albuminuria and CKD in patients with T2DM.

Methods: Data from 01/2013-12/2017 period were obtained from the Population Health Data Archive's Diabetes Complications Data Set. A total of 1048 patients with T2DM were included in this study. CKD is defined as an estimated glomerular filtration rate < 60 ml/min-1.1.73 m-2 or a urinary albumin-to-creatinine ratio (UACR) ≥ 30 mg/g. Albuminuria is defined as a UACR ≥ 30 mg/g. The TyG index is calculated by measuring the triglyceride and fasting blood glucose levels. Logistic regression models were used to analyze the association between albuminuria, CKD with T2DM and TyG index.

Results: We identified 1048 subjects, 63.03% of whom were men. The mean age was 46.21 years, and the mean body mass index was 26.742 kg/m2. CKD and albuminuria detection rates showed an increasing trend in the different TyG subgroups. (p = 0.008, p = 0.006). Using the Q1 group as a baseline, the risk of albuminuria and CKD was significantly greater in the group Q3 (OR = 1.514, 95% CI 1.121-2.047 P = 0.05), and the same result was obtained after adjusting for covariates (OR = 2.241, 95% CI 1.245-4.034, P = 0.007). Subgroup analyses revealed a significant increase in the incidence of albuminuria and CKD in the group Q3 compared to that in the Q1 group. 

Conclusions: The TyG index is positively associated with albuminuria and CKD in patients with T2DM and may be a marker for predicting the occurrence of early kidney injury in patients with T2DM. Clinicians should test this indicator early to detect lesions and improve patient prognosis.

Comment 5 The whole manuscript sections (introduction, method, result, discussion) should be revised. The manuscript does not follow a standard structure of manuscript writing. I recommend a native English-speaking revision to address the many stylistic issues, the manuscript lacks fluency and the English in the text is EXTREMELY poor, complete revision is necessary. these are some examples of the text that should be revised:

Response: Thank you very much for your suggestions, we have carefully revised the objectives, methods, results, and conclusions of the abstract according to your suggestions, and we hope that our revisions will make you have a satisfactory reading experience!

Introduction

Comment 6 The results concerning hard end-points should be stated. In my opinion, it is too brief and not focused on clinical aspects clarifying the specific factors, tests, or markers evaluated and their significance in understanding the pathological roles in patients with CKD. These are some of the many grammatical, stylistic, and spelling errors: “Chronic kidney disease (CKD) is a microvascular complication of diabetes mellitus (DM) (1). As the prevalence of CKD increases, so does the risk of cardiovascular disease, end-stage renal disease (ESRD), and cumulative mortality (2)” The whole paragraph should be revised.

Response: Thank you for your suggestion, we have rewritten the introduction section and have invited native English speakers to correct the grammatical, stylistic, and spelling errors. I have enriched the introduction with a description of the Tyg index to the clinic, which includes specific factors for its assessment and specific applications and explains the significance of the Tyg index in terms of its pathological role in patients with CKD. You can see the overall changes to the introduction in the revised manuscript and in the response to comment 9

Comment 7: Paraphrase "CKD WAS defined as estimated glomerular filtration …" to "CKD is defined as estimated glomerular filtration…"

Response: Thank you for your suggestion, we have changed it to “CKD is defined as estimated glomerular filtration…"

Comment 8: some well-defined and basic scientific facts are mentioned as scientific suggestions for example: “It has been suggested that insulin resistance (IR) is closely associated with an increased risk of kidney disease in diabetic populations”

Response: Thank you for your suggestion, we have rewritten this section and avoided the phenomenon you mentioned.

Comment 9: There should be an explanation regarding the definition of the TyG index, its utility, and previous research on this index. Delve into the concept of clinical importance and the superiority of using this index in CKD. Overall, significant revisions are required for structural, grammatical, and clarity issues. Additionally, the clinical implications of the findings need further discussion.

Response: Thank you for your suggestions, I have revised the introduction section as a whole based on your comments. The TyG index is a combination of FPG and TG. The specific formula is ln [FPG (mg/dl) *TG (mg/dl)], we discuss the previous research on this index and also describe its superiority and reliability compared to HOMA-IR and HIEC tests for renal disease, and discuss and summarize the clinical importance of its use.The overall introduction to introduction can be seen in the revised manuscript marker red and below.

Introduction

Type 2 diabetes mellitus (T2DM) is the most common metabolic disorder that leads to numerou

---

## [Decision Letter · Decision Letter 1]

6 Sep 2024

PONE-D-24-07664R1Association of TyG index with albuminuria and Chronic Kidney Disease with Type 2 Diabetes MellitusPLOS ONE

Dear Dr. Wang,

Thank you for submitting your manuscript to PLOS ONE. After careful consideration, we feel that it has merit but does not fully meet PLOS ONE’s publication criteria as it currently stands. Therefore, we invite you to submit a revised version of the manuscript that addresses the points raised during the review process.

We look forward to receiving your revised manuscript.

Kind regards,

Amirmohammad Khalaji

Academic Editor

PLOS ONE

Reviewers' comments:

Reviewer's Responses to Questions

**Comments to the Author**

1. If the authors have adequately addressed your comments raised in a previous round of review and you feel that this manuscript is now acceptable for publication, you may indicate that here to bypass the “Comments to the Author” section, enter your conflict of interest statement in the “Confidential to Editor” section, and submit your "Accept" recommendation.

Reviewer #1: All comments have been addressed

Reviewer #2: All comments have been addressed

Reviewer #3: All comments have been addressed

2. Is the manuscript technically sound, and do the data support the conclusions?

Reviewer #1: No

Reviewer #2: Yes

Reviewer #3: Yes

3. Has the statistical analysis been performed appropriately and rigorously? 

Reviewer #1: I Don't Know

Reviewer #2: Yes

Reviewer #3: Yes

4. Have the authors made all data underlying the findings in their manuscript fully available?

Reviewer #1: No

Reviewer #2: No

Reviewer #3: (No Response)

5. Is the manuscript presented in an intelligible fashion and written in standard English?

Reviewer #1: No

Reviewer #2: Yes

Reviewer #3: (No Response)

6. Review Comments to the Author

**Reviewer #1:** (No Response)

**Reviewer #2:** The authors have addressed all the comments by this reviewer and the manuscript is acceptable in its current form.

**Reviewer #3:** Reviewer Comments

Li and Wang examined the connection between the TyG index, albuminuria, and CKD in individuals with T2DM. Results indicated a positive association between the TyG index and both albuminuria and CKD, with higher odds ratios observed for those in the highest TyG index quartile. The research topic is interesting and valuable. The revised version addresses all the problems in the first version of the manuscript; the language fluency and punctuation are proper, the article's structure is good, and the discussion section now has an appropriate elaboration of the findings in the context of relative literature.

Language and Writing:

1. The revised manuscript has proper English fluency and punctuation.

Introduction:

1. The introduction is technically firm and informative.

Method:

1. The inclusion and exclusion criteria are missing.

2. The eGFR formula is mistyped. (Line 104)

Results:

1. The results is technically firm and informative.

Discussion:

1. The discussion now has an appropriate elaboration of the findings in the context of relative literature and other studies.

7. PLOS authors have the option to publish the peer review history of their article (what does this mean?). If published, this will include your full peer review and any attached files.

Reviewer #1: No

Reviewer #2: No

Reviewer #3: No

---

## [Author Response · Author response to Decision Letter 1]

7 Sep 2024

Reviewer #1:

Comment 1: Is the manuscript technically sound, and do the data support the conclusions? Reviewer #1: No

Response：Thank you for your detailed review and valuable feedback on our manuscript. Your comments have been instrumental in improving the quality of our article. We understand your concerns regarding the data source and the conclusion. The data come from the Diabetes Complications Data Set which were collected by the General Hospital of the People's Liberation Army of China from 01/2013 to 12/2017 and it was stored in the Population Health Data Archive. The website is https://www.ncmi.cn/phda/dataDetails.do?id=CSTR:A0006.11.A0005.201905.000282(Language is Chinese). The dataset contains 87 variables for 3,000 patients with T2DM. For our analysis, after passing the inclusion and exclusion criteria, a total of 1048 person was included in the study. The data can be obtained by any approved requester and used for verification. In addition, the General Hospital of the People's Liberation Army of China, located in the capital city of Beijing, is a premier national-level tertiary hospital. It serves a diverse patient population from across the country, which contributes to the reliability and generalizability of our results.

Comment 2: Has the statistical analysis been performed appropriately and rigorously? 

Reviewer #1: I Don't Know.

Response: Thank you very much for your valuable suggestions. We have refined and clarified our data analysis methods as follows:

1)First, we performed normality tests on the data. For normally distributed data, we present results as mean ± standard deviation, while non-normally distributed data are presented as medians (interquartile ranges). Continuous variables were compared using t-tests or Mann-Whitney U tests, and categorical variables were described with frequencies or percentages, with chi-square tests used for inter-group comparisons. These details are included in Table 1 of the revised manuscript.

2) To further investigate the relationship between the TyG index and albuminuria and chronic kidney disease (CKD), we categorized the TyG index into three groups (Q1, Q2, Q3) and performed binary logistic regression analysis. We adjusted for potential confounders such as hypertension, fatty liver, hyperlipidemia, coronary heart disease, total cholesterol, glycosylated hemoglobin, sex, age, fasting glucose, retinopathy, and eGFR. The results demonstrated that the TyG index is an independent risk factor for albuminuria and CKD in T2DM patients. Detailed results can be found in Table 3 and Table 5.

3) Further adjustments for covariates and stratified analyses of variables were performed to validate the results, and the results were consistent with the primary analysis, reinforcing the TyG index as an independent risk factor. Detailed results can be seen in Table 4 and Table 6.

Comment 3: Have the authors made all data underlying the findings in their manuscript fully available?Reviewer #1: No

Response: Thank you for your patient work, our data comes from publicly available databases. The example data from the Diabetes Complications Data Set can be acquired and was uploaded as S1_Table. Once the application has been approved, all the data can be viewed and used. The website on which the data is obtained is https://www.ncmi.cn/phda/dataDetails.do?id=CSTR:A0006.11.A0005.201905.000282.

Comment 4: Is the manuscript presented in an intelligible fashion and written in standard English? Reviewer #1: No.

Response: Thank you for your meticulous work. To enhance the quality of the manuscript, we have had it reviewed and edited by a native English speaker to ensure clarity and precision in language.

Reviewer #2:

Comment 1: Have the authors made all data underlying the findings in their manuscript fully available?Reviewer #2: No

Response: Thank you for your advice, our data comes from publicly available databases. The example data from the Diabetes Complications Data Set can be acquired and was uploaded as S1_Table. Once the application has been approved, all the data can be viewed and used. The website on which the data is obtained is https://www.ncmi.cn/phda/dataDetails.do?id=CSTR:A0006.11.A0005.201905.000282.

Comment 2: The authors have addressed all the comments by this reviewer and the manuscript is acceptable in its current form.

Response: Thanks for your extremely patient work. I couldn't have improved my dissertation without your very instructive advice, thank you for your work on my paper and I wish you all the best.

Reviewer #3: 

Comment 1: Li and Wang examined the connection between the TyG index, albuminuria, and CKD in individuals with T2DM. Results indicated a positive association between the TyG index and both albuminuria and CKD, with higher odds ratios observed for those in the highest TyG index quartile. The research topic is interesting and valuable. The revised version addresses all the problems in the first version of the manuscript; the language fluency and punctuation are proper, the article's structure is good, and the discussion section now has an appropriate elaboration of the findings in the context of relative literature. Language and Writing:1. The revised manuscript has proper English fluency and punctuation. The introduction is technically firm and informative.

Response: Thank you for your acknowledgement of my paper. I really can't improve my paper without your valuable comments and thank you for the time you have given to my paper.

Comment 2: Method:1. The inclusion and exclusion criteria are missing.

Response: Thank you for the reminder, I apologize for such an error and I added the inclusion and exclusion criteria. And it is identified in revised the manuscript by the application of red font (line 88-92).

Comment 3: The eGFR formula is mistyped. (Line 104)

Response: Thank you for your suggestions. I have made the necessary revision to the formula and highlighted the changes in red in revised the manuscript(line108-109).

Comment 4: The results is technically firm and informative. The discussion now has an appropriate elaboration of the findings in the context of relative literature and other studies.

Response: Thank you for your patient work, I appreciate your suggestive tips to improve my paper.

Once again, we would like to thank the editor and all reviewers for their valuable comments and suggestions. Such comments are valuable and helpful for our revision and improvement. We sincerely hope that these revisions meet your expectations and that the manuscript can be considered for publication in PLOS ONE. Should you have any further questions or require additional modifications, please do not hesitate to contact us. We are committed to making any necessary revisions and improvements.

---

## [Decision Letter · Decision Letter 2]

7 Oct 2024

Associations of the TyG index with albuminuria and chronic kidney disease in patients with type 2 diabetes mellitus

PONE-D-24-07664R2

Dear Dr. Wang,

We’re pleased to inform you that your manuscript has been judged scientifically suitable for publication and will be formally accepted for publication once it meets all outstanding technical requirements.

Kind regards,

Amirmohammad Khalaji

Academic Editor

PLOS ONE

Additional Editor Comments (optional):

Reviewers' comments:

Reviewer's Responses to Questions

**Comments to the Author**

1. If the authors have adequately addressed your comments raised in a previous round of review and you feel that this manuscript is now acceptable for publication, you may indicate that here to bypass the “Comments to the Author” section, enter your conflict of interest statement in the “Confidential to Editor” section, and submit your "Accept" recommendation.

Reviewer #1: All comments have been addressed

Reviewer #3: All comments have been addressed

2. Is the manuscript technically sound, and do the data support the conclusions?

Reviewer #1: Yes

Reviewer #3: Yes

3. Has the statistical analysis been performed appropriately and rigorously? 

Reviewer #1: Yes

Reviewer #3: Yes

4. Have the authors made all data underlying the findings in their manuscript fully available?

Reviewer #1: Yes

Reviewer #3: Yes

5. Is the manuscript presented in an intelligible fashion and written in standard English?

Reviewer #1: Yes

Reviewer #3: Yes

6. Review Comments to the Author

Reviewer #1: (No Response)

Reviewer #3: (No Response)

7. PLOS authors have the option to publish the peer review history of their article (what does this mean?). If published, this will include your full peer review and any attached files.

Reviewer #1: No

Reviewer #3: No

---

## [Editor Report · Acceptance letter]

17 Oct 2024

PONE-D-24-07664R2 

PLOS ONE

Dear Dr. Wang, 

I'm pleased to inform you that your manuscript has been deemed suitable for publication in PLOS ONE. Congratulations! Your manuscript is now being handed over to our production team.

Kind regards, 

on behalf of

Dr. Amirmohammad Khalaji 

Academic Editor

PLOS ONE